# Dosimetric outcomes of preoperative treatment planning with intraoperative optimization using stranded seeds in prostate brachytherapy

**Jason Joon Bock Lee[1,2☯], Eungman Lee[1,3☯], Won Hoon Choi[4], Jihun Kim[1], Kyung Hwan Chang[5], Dong Wook Kim[1], Han Back Shin[1], Tae Hyung Kim[1,6], Hwa Kyung Byun[1], Jaeho Cho[1]***

1 Department of Radiation Oncology, Yonsei Cancer Center, Yonsei University College of Medicine, Seoul, South Korea, 2 Department of Radiation Oncology, Kangbuk Samsung Hospital, Sungkyunkwan University School of Medicine, Seoul, South Korea, 3 Department of Radiation Oncology, Ewha Womans University College of Medicine, Seoul, South Korea, 4 INFINITT Healthcare, Seoul, South Korea, 5 Department of Digital Health Solution, Douzone Bizon, Seoul, South Korea, 6 Department of Radiation Oncology, Nowon Eulji Medical Center, Seoul, South Korea

☯ These authors contributed equally to this work.
* jjhmd@yuhs.ac

**Data Availability Statement:** For restrictions on sharing data publicly, data cannot be shared publicly because of potentially identifying or sensitive patient information. Data are available

## Abstract

This study aimed to evaluate the quality of low-dose-rate (LDR) prostate brachytherapy (BT) based on treatment-related dosimetric outcomes. Data of 100 patients treated using LDR BT with stranded seeds from November 2012 to November 2017 were collected. The prescription dose for the prostate was 145 Gy. The dose constraints for the preoperative plan were: V100% $\geq$ 95%, V150% $\leq$ 60%, V200% $\leq$ 20% for the prostate; V100% for rectum, $\leq$ 1 cc; and V200 Gy for urethra, 0.0 cc. Intraoperative real-time dose calculation and postoperative dose distribution analysis on days 0 and 30 were performed. Median dosimetric outcomes on days 0 and 30 respective were: V100% 92.28% and 92.23%, V200% 18.63% and 25.02%, and D90% 150.88 Gy and 151.46 Gy for the prostate; V100% for the rectum, 0.11 cc and 0.22 cc; and V200 Gy for the urethra, 0.00 cc and 0.00 cc, respectively. Twenty patients underwent additional seed implantation to compensate for insufficient dose coverage of the prostate. No loss or substantial migration of seeds or severe toxicity was reported. With stranded seed implantation and intraoperative optimization, appropriate dose delivery to the prostate without excessive dose to the organs at risk could be achieved.

## Introduction

Low-dose-rate (LDR) prostate brachytherapy (BT) is one of the most effective treatment modalities for early-stage prostate cancer, along with other therapies, such as surgery and external beam radiotherapy (EBRT). BT is associated with excellent cancer-related outcomes and results in remarkable improvement in quality of life, including reduction of urinary incontinence and sexual dysfunction, compared to other radical treatment strategies [1–3].

from our Institutional Review Board for researchers who meet the criteria for access to confidential data:jbwolf86@gmail.com.

**Funding:** The author(s) received no specific funding for this work.

**Competing interests:** The authors have declared that no competing interests exist.

However, earlier generations of LDR BT had a potential risk of severe complications related to seed migration [4,5]; therefore, in the modern-era LDR BT, stranded seeds are commonly used to avoid seed migration after implantation [6,7]. The dose to the urethra and rectum is another important aspect of modern high-quality LDR BT, as risks of urinary complications and radiation proctitis do exist, although LDR BT could be safely performed with minimal short- and long-term toxicities in most cases [8]. A previous study reported the use of metal-oxide–semiconductor field-effect transistor (MOSFET) in in vivo dosimetry and quality assurance for BT [9]. The radiation dose delivery of LDR BT is relatively more effective than that of EBRT, as it can heterogeneously create an intensely irradiated area with increased probability for tumor cell-killing and local tumor control [10].

Unlike earlier generations of BT, fourth generation BT with stranded seeds [11] and intraoperative optimization enables a more effective radiation dose delivery with lower risk of seed migration or treatment-related side effects. Various intraoperative circumstances may prevent the implantation of seeds as planned; therefore, intraoperative dose monitoring and real-time optimization are essential for quality-controlled implantation. Given that BT requires appropriate technical procedures, we evaluated those in detail with a description in intraoperative optimization and MOSFET utilization. Furthermore, in this study, we reported the dosimetric outcomes of LDR BT with optimized implantation and the results of the postoperative evaluation on day 0 and day 30.

## Materials and methods

### Patients

Medical records of 100 patients with early-stage prostate cancer, who underwent LDR BT between November 2012 and November 2017, were reviewed. Based on the Memorial Sloan Kettering Cancer Center (MSKCC) risk grouping criteria, patients with low-risk [initial prostate-specific antigen (PSA) $\leq$ 10.0 ng/mL, Gleason score (GS): 2–6, and Stage T1a–T2b)] and intermediate-risk (initial PSA > 10.0 ng/mL or GS $\geq$ 7 or Stage $\geq$ T2c; 1 risk facor) prostate cancer were indicated to undergo LDR BT [12]. Further, patients with an international prostate symptom score $\leq$ 20 were considered as the preferred candidates for BT.

### Preoperative processes

Fig 1 shows the entire process of LDR BT from preoperative simulation to postoperative evaluation. Identification of pubic arch interference in a lithotomy position was important for determining the candidates for LDR BT. As a large prostate volume is more likely to cause pubic arch interference, patients with prostate volume smaller than 50 cc were considered appropriate candidates for BT. Preoperatively, magnetic resonance (MR) images of each patient were acquired and transferred to MIM software version 6.4.6 (MIM Software Inc., Cleveland, OH, USA), and the prostate volume was measured based on these images. These MR images were used to determine whether the patient is eligible for BT. In patients with an initial prostate volume larger than 50 cc, neoadjuvant androgen deprivation therapy was performed before BT to reduce the volume below 50 cc. Transrectal ultrasound (TRUS) images were acquired with a biplane probe (Type 8848) connected to a ProFocus2202 US scanner (BK Medical ApS, Herlev, Denmark) during preoperative simulation. Each patient underwent preoperative treatment planning based on these TRUS images. These images were also used to predict pubic arch interference, which prevents proper needle insertion into the peripheral zone of the prostate. Preoperative simulation was performed in a supine lithotomy position, with the Foley catheter inserted. Prostate and adjacent organs at risk (OAR), including the urethra, bladder, rectum, and seminal vesicle, were contoured by experienced physicians using

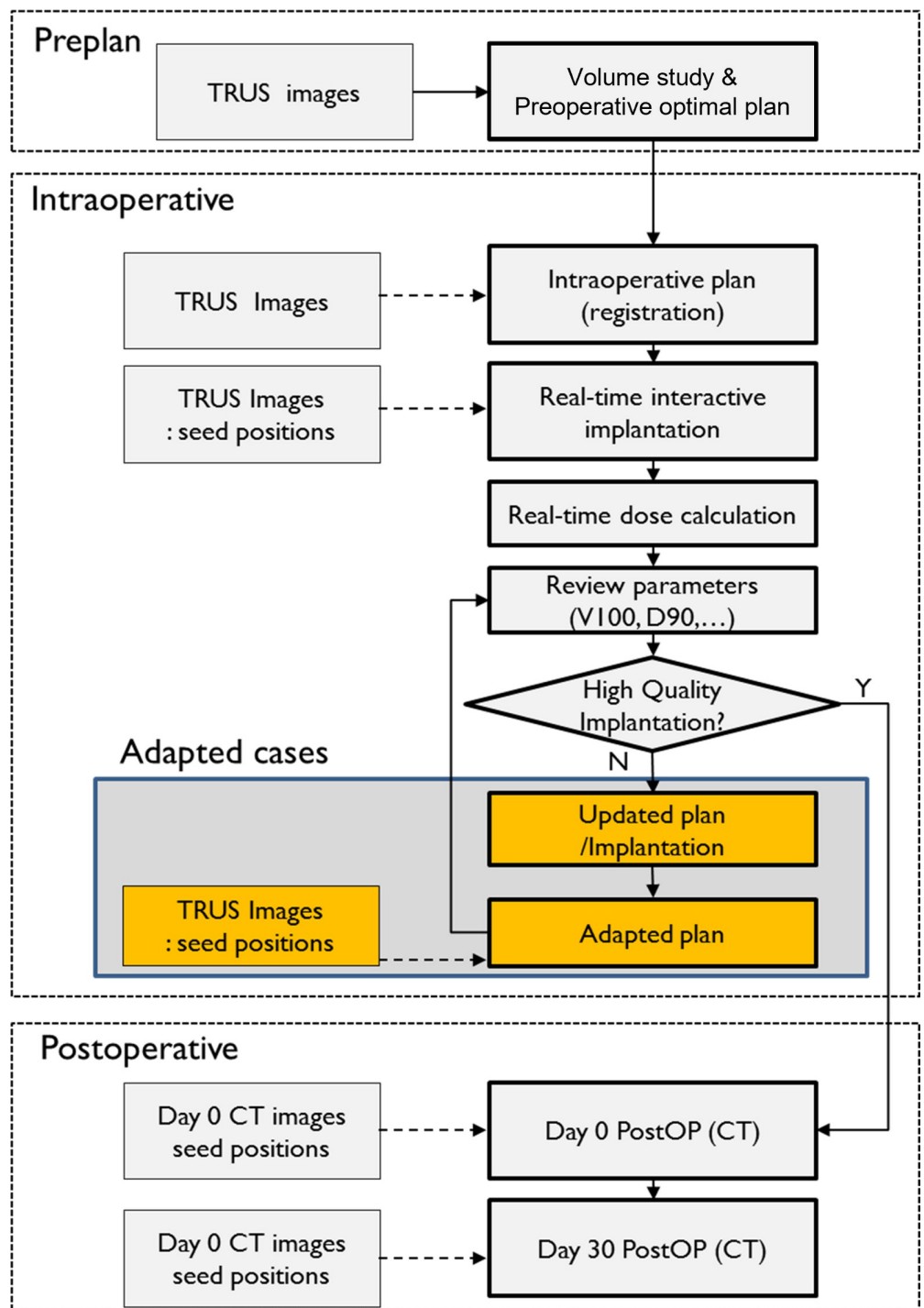

**Fig 1. The schematics describe the procedure of low-dose-rate prostate brachytherapy.** First, preoperative planning is conducted based on the transrectal ultrasound (TRUS) images. Intraoperatively, the TRUS images are acquired to verify the reproducibility of the prostate, urethra, bladder, and rectum between the preplan and intraoperative procedure and to monitor the needle and the seeds. After seed implantation, TRUS-based dosimetry is performed. When the target coverage was not sufficient, intraoperative optimization was conducted immediately. On postoperative day 0, dosimetry was conducted using computed tomography images. Further, on postoperative day 30, dosimetry was conducted for monitoring seed displacement.

VariSeed software version 8.0.1 (Varian Medical Systems, Palo Alto, CA, USA). The prescribed dose for the prostate was 145 Gy. Dose constraints for critical OARs, including the prostate, were as follows: V100% for the prostate, > 95%; V150% for the prostate, ≤ 60%; V200% for the prostate, ≤ 20%; V100% for the rectum, < 1 cc; and V200 Gy for the urethra, 0.0 cc.

## Surgical procedures and intraoperative optimization

Patients received seed implantation through a modified peripheral loading technique to avoid urethral injury [13,14] in a supine lithotomy position identical to the simulation. Stranded iodine-125 seeds (Best Medical International Inc., Springfield, VA, USA) were used in all patients. In 50 patients, a MOSFET (Linear 5ive MOSFET Array Dosimeter, TN-502LA5, Best Medical Canada, ON, Canada) was inserted into the Foley catheter to measure the radiation dose to the urethra and confirm whether the actual urethral dose met the prescription criteria. It was positioned inside the Foley catheter such that the first probe was located in the bladder and the second probe was located at the bladder neck. The schematic description of the relative location of MOSFET, TRUS, and inserted seeds is shown in Fig 2. Two-minute in-vivo MOS-FET measurements were performed intraoperatively, and 2-minute and 10-minute measurements were acquired at the end of the surgical procedure. The electrical bias measured by the MOSFET detectors was converted using a vendor-provided calibration factor (15.2 mV/cGy). These measured doses were converted to the life-time radiation doses, which the patient would receive during the whole decaying period of the implanted seeds. The maximum value of the 10-minute measurement results was compared to the maximum urethral dose in the preoperative plan, further assuring the dosimetric quality of the procedure. Intraoperative TRUS images were acquired before the actual implantation began. These intraoperative images were then compared with corresponding images obtained during preoperative simulation on monitors installed in the operating room to determine the necessity of intraoperative improvisation of needle insertion. The prostate and OARs remained consistent throughout the period between the simulation and operation; therefore, contours had to be delineated again only in rare cases. The real-time estimated radiation dose to the prostate and other OARs was

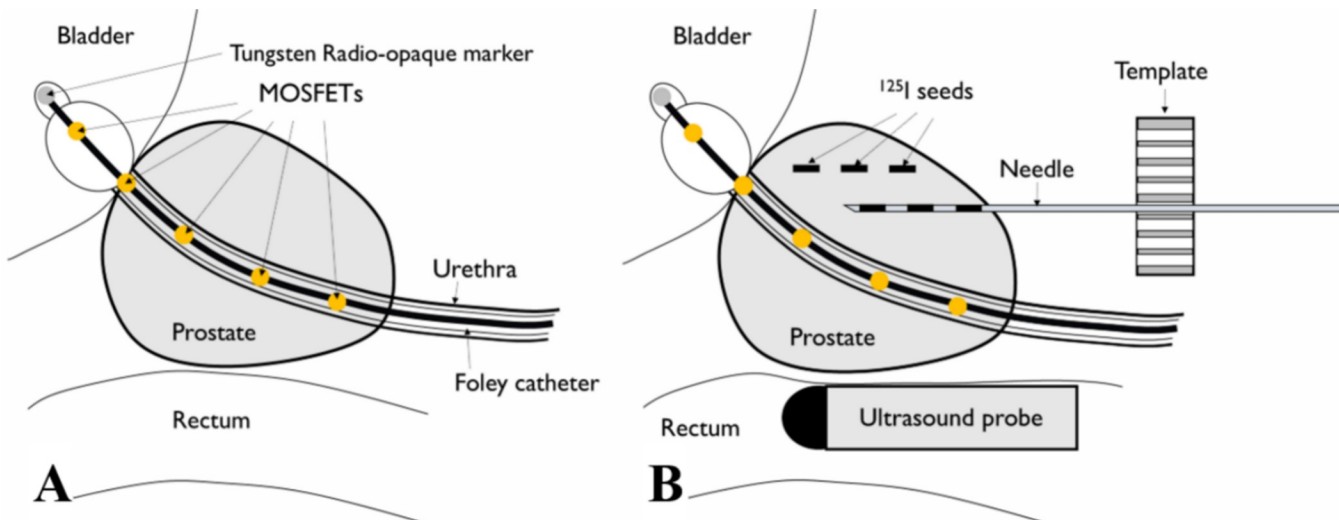

**Fig 2. Schematic illustration of the relative location of transrectal ultrasound (TRUS), metal-oxide–semiconductor field-effect transistor (MOSFET), and implanted seeds during the low-dose-rate prostate brachytherapy procedure.** (A) MOSFET is inserted through a Foley catheter, and the second probe is usually located at the bladder neck. (B) The needle containing radioactive seeds goes through the template under real-time guidance based on TRUS images. After the needles are removed, the radioactive seeds remain implanted in the prostate.

contemporaneously calculated by VariSeed software version 8.0.1., based on the TRUS images and seed locations acquired during the operation. Results of these analyses were compared with the preoperative plan to determine whether the actual implanted seeds could ultimately fulfill the abovementioned dose parameters. When the dose distribution did not match that of the preoperative plan due to various intraoperative factors, the implant was optimized in the operating room, and additional seeds were inserted to ensure sufficient target coverage, while also ensuring that the prostate V100% was > 95%, the rectum V100% was < 1 cc, and the urethra V200 Gy was 0 cc. Therefore, a total of four spare needles, two needles with two extra seeds and two needles with three extra seeds, were prepared in case of target underdose for each patient. A kidney, ureter, and bladder radiograph was acquired after entire operation to observe the gross distribution of the implanted seeds in the prostate area.

### Postoperative dosimetry

CT-based postoperative dosimetry was evaluated using CT images acquired for both Day 0 (immediately after the procedure) and Day 30. All postoperative CT images were obtained with a slice thickness of 1 mm for appropriate visualization of the implanted seeds. The detailed process was as follows. First, on the CT image, VariSeed software version 8.0.1. automatically localized the implanted seeds by thresholding CT numbers, and the physician confirmed the property of identified seeds locations. Second, all of the contours generated by TRUS, except the rectum, were rigidly aligned to the postoperative CT images. The rectum was manually contoured regarding its substantial geometrical change in the CT image compared to the TRUS image. Finally, the radiation dose to the prostate and OARs was calculated with the localized brachytherapy seeds. All of the computational dosimetries, including preoperative, intraoperative, and postoperative dosimetry, were performed using VariSeed 8.0.1. Specifically, radiation doses were calculated using a point source model, based on the updated American Association of Physicists in Medicine (AAPM) TG-43 formalism. The air-kerma strength of the I-125 source was set to 0.391 mCi.

### Follow-up and statistics

CT scanning was performed 1 month postoperatively to evaluate whether the implanted seeds remained within the prostate without seed migration. The dose distribution was also calculated based on this CT scan. All patients were followed up through consecutive outpatient clinic visits with serial PSA and toxicity evaluation. Toxicities during the follow-up period were evaluated based on the Common Terminology Criteria for Adverse Events Version 4.03. All statistical analyses in this study were performed using IBM SPSS Statistics for Windows, Version 24.0 (IBM Corp., Armonk, NY, USA).

### Institutional review board statement

Ethical approval for this study was obtained from the Institutional Review Board of Yonsei University Health System, Severance Hospital, South Korea (Approval No.: 4-2019-0767).

All subjects had provided written permission for their medical records to be used for research purposes as provided for.

## Results

### Patient characteristics

The patient characteristics are summarized in Table 1. With a median age of 64 years, the T stages of 99 patients were found to be < 2c, and 87 patients had a GS < 7. The median initial

**Table 1. Patient characteristics.**

| Patient Characteristic | Median (range) /No. of patients (%) |
|---|---|
| Age (yrs) | 64 (46–82) |
| T stage | |
| 1c | 26 (26.0) |
| 2a | 45 (45.0) |
| 2b | 20 (20.0) |
| 2c | 8 (8.0) |
| 3a | 1 (1.0) |
| Gleason score | |
| 6 (3+3) | 55 (55.0) |
| 7 (3+4) | 21 (21.0) |
| 7 (4+3) | 13 (13.0) |
| 8 (4+4) | 11 (11.0) |
| Initial PSA$ | 7.39 (2.82–90.64) |
| MSKCC$ risk group | |
| Low | 44 (44.0) |
| Intermediate | 50 (50.0) |
| High | 6 (6.0) |
| IPSS* score | 10 (0–34) |
| No. of total biopsy | 12 (6–16) |
| No. of positive biopsy | 2 (1–12) |
| No. of patients with hormonal therapy before BT∀ | 18 (18.0) |

$ Prostate specific antigen.

§ Memorial Sloan Kettering Cancer Center.

* International Prostate Symptom Score.

∀ Brachytherapy.

PSA was 7.39 ng/mL, and 92 patients were classified into the low–intermediate risk group according to the MSKCC risk grouping criteria. The median follow-up period was 42 months (range: 24–93 months). Seventeen patients received neoadjuvant androgen deprivation therapy to reduce the prostate volume.

## Dosimetric characteristics

The dosimetric characteristics of the patients are summarized in Table 2. The median number of total implanted needles and seeds were 20 and 76, respectively. Intraoperatively, radioactive seeds were implanted both within the prostate and around the peripheral region of the prostate to effectively cover the seminal vesicle. The median prostate volume was 28.68 cc (range: 10.39–48.01 cc). Although the dosimetric criteria were met in almost all preoperative BT plans, in rare cases, postoperative dosimetry on days 0 and 30 showed a slight violation of these criteria. For instance, on day 0, 26 patients had V100% for the prostate, < 90.0%, while one patient had a V100% of 76.63%. To confirm this observation, D90% of the prostate, i.e., a dose covering 90% of the prostate volume, was also evaluated; the median D90% was 150.88 Gy (range: 131.02–174.89 Gy). Rectal and urethral dose constraints were strictly imposed, and these criteria were met in almost all postoperative dosimetric analyses. The number of seeds implanted increased proportionally with the prostate volume, as more seeds were required to apply a sufficient radiation dose to larger prostates. Approximately one additional seed was needed to cover a 2-cc increment of prostate volume. Fig 3 illustrates the linear correlation between the prostate

**Table 2. Dosimetric characteristics.**

| Dosimetric characteristics | Median (range) |
|---|---|
| No. of needles implanted | 20 (14–33) |
| No. of seeds used | 76 (52–102) |
| *Preoperative dosimetry* | |
| Prostate volume (cc) | 28.68 (10.39–48.01) |
| Prostate $V_{100\%}$ (%) | 95.72 (91.33–98.48) |
| Prostate $V_{200\%}$ (%) | 18.63 (9.39–34.73) |
| Prostate $D_{90\%}$ (Gy) | 159.64 (148.26–172.06) |
| Rectum $V_{100\%}$ (cc) | 0.16 (0.00–0.67) |
| Urethra $V_{200\ Gy}$ (cc) | 0.00 (0.00–0.00) |
| *Postoperative dosimetry—Day 0* | |
| Prostate $V_{100\%}$ (%) | 92.28 (76.63–97.81) |
| Prostate $V_{200\%}$ (%) | 18.63 (9.39–34.73) |
| Prostate $D_{90\%}$ (Gy) | 150.88 (131.02–174.89) |
| Rectum $V_{100\%}$ (cc) | 0.11 (0.00–1.90) |
| Urethra $V_{200\ Gy}$ (cc) | 0.00 (0.00–0.03) |
| *Postoperative dosimetry—Day 30* | |
| Prostate $V_{100\%}$ (%) | 92.23 (80.50–100.00) |
| Prostate $V_{200\%}$ (%) | 25.02 (10.77–81.99) |
| Prostate $D_{90\%}$ (Gy) | 151.46 (126.00–181.35) |
| Rectum $V_{100\%}$ (cc) | 0.22 (0.00–1.70) |
| Urethra $V_{200\ Gy}$ (cc) | 0.00 (0.00–0.26) |

volume and number of seeds implanted [or total activity (mCi)] with an $R^2$ value of 0.71. Only two of the 50 patients whose in vivo MOSFET data were available showed a maximum urethral dose > 200 Gy, while the median urethral maximum dose was 152.8 Gy (range: 113.23–486.53 Gy). However, the maximum dose recorded was 486.53 Gy in one of the patients, but the dose measured at the other four points ranged between 0.00 and 35.35 Gy, suggesting a strong possibility of measurement error. The maximum dose recorded in another patient was 211.35 Gy.

## Intraoperative dynamic dose optimization

Intraoperative dynamic dose optimization using TRUS images, which show the location of the implanted seeds, allows comparison between actual real-time dose distribution and the preoperative plan. Twenty patients with V100% < 95% in intraoperative dose calculation underwent successful extra seed implantation without compromising the dose constraints on the OARs. The linear correlation of total activity between the preoperative plan and intraoperative optimization in patients who required additional radioactive seeds inserted are presented in S1 Fig. The most important indicators used in this study to assess the treatment quality of BT were the V100% and D90% of the prostate. At the end of the implantation, if the intraoperative dose optimization result showed that the V100% of the prostate was < 95% or D90% was < 140 Gy, this would be considered insufficient target coverage, and additional seeds were inserted for compensation. In 20 patients, 2–5 additional stranded seeds were inserted. Underdosed areas usually occurred around the central zone proximal to the urethra because of cautious implantation to avoid toxicities, and 15 patients received additional seed insertion for this reason. For the other five patients, additional implantation was located around the peripheral area of the prostate to compensate for substantial swelling of the prostate, thereby covering the adjacent periprostatic fatty tissue. Fig 4 shows an example of additional seed implantation

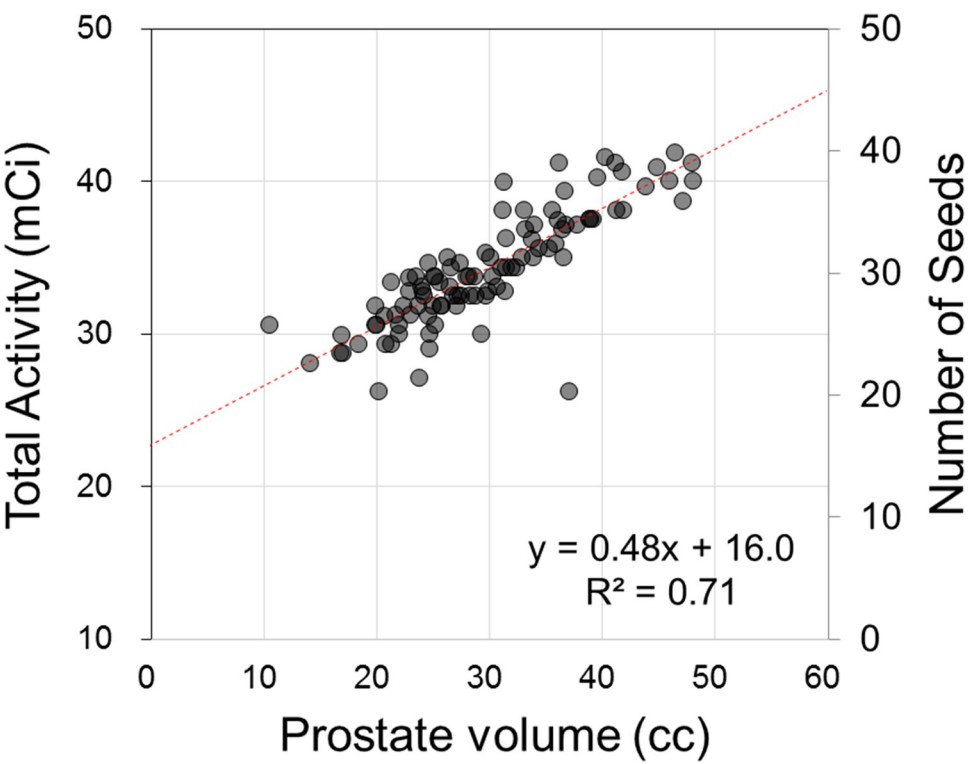

**Fig 3. Low-dose-rate brachytherapy nomogram.** Data are fitted to a linear expression. In most cases, the total activity or number of seeds is proportional to the prostate volume.

in the central portion of the prostate while maintaining the dose constraints for OARs with improved prostate dose coverage. As a result of this optimization, the dosimetric outcome improved (Fig 5A). Fig 5B and 5C present the reduction of the difference between the dose distribution in the preoperative simulation and the intraoperative dose calculation by additional compensatory implantation. These figure panels show that intraoperative dynamic optimization can improve the dosimetric outcomes of LDR BT.

## Comparison of dosimetric outcomes at day 0 and day 30

At the first follow-up, 30 days after BT, another CT scan was performed to evaluate the dose distribution and examine the possibility of seed migration. Fig 6 illustrates examples of the seed placement pattern at representative sections of the prostate at each time point of BT. The intraoperative placement pattern was determined based on TRUS images, and the postoperative placement patterns were depicted on CT images. All seeds were implanted within a 4-cm range from the urethra, suggesting that precise seed implantation was needed to avoid complications (Fig 6A). Seed placement in a model patient is shown in panels Fig 6B–6D using the transverse, coronal, and sagittal views of a representative slice of images acquired with intraoperative TRUS and immediately postoperatively and 30 days postoperatively on CT. No evidence of substantial seed migration was found during these examinations in any patient. Dosimetric comparison was made between day 0 and day 30 in terms of the dose to the prostate and OARs. The D90% and V100% of the prostate on days 0 and 30 after seed implantation are displayed in Fig 7; the dose coverage to the prostate was quite well-maintained postoperatively. The difference between day 0 and day 30 in the V200 Gy of the urethra and V100% of the rectum is illustrated in Fig 8. The difference ranged from −0.2 cc to 0.11 cc in the V200 Gy

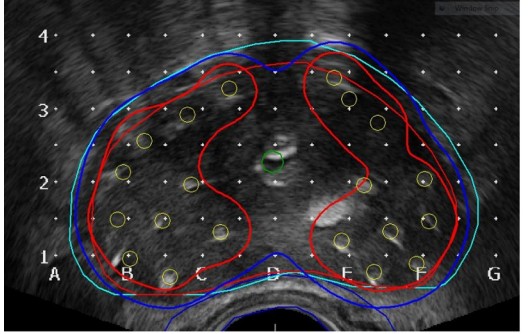
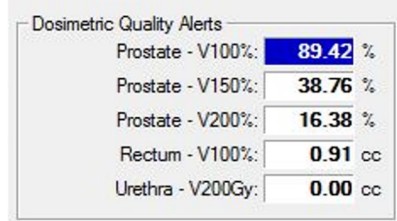

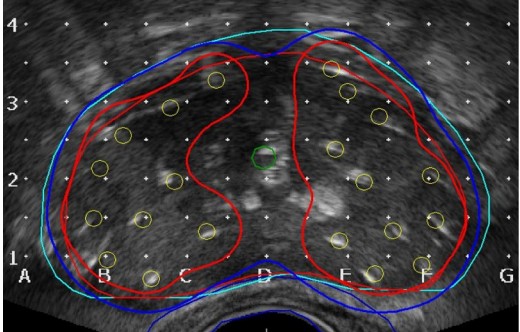

**Fig 4. An exemplary case of cold-spot reduction around the urethra and improvement of dosimetric parameters.**
Red lines are isodose lines for prostate V100%, orange circles indicate the actual implanted seed locations, and real-time dose distribution was calculated instantaneously according to these locations. With real-time intraoperative optimization, both better dose coverage and maintained dose constraints were possible.

of the urethra and −1.49 cc to 1.28 cc in the V100% of the rectum. The dose to the OARs remained stable without substantial change in all patients.

## Early clinical outcomes

No intraprostatic gross failure or cancer-related mortality was reported during the follow-up period, with a 3-year biochemical failure (BCF)-free survival of 98.4%. However, one regional failure and two BCF cases were noted. A patient who experienced regional failure with solitary obturator lymph-node metastasis was subjected to stereotactic body radiotherapy on the metastatic lymph node, with subsequent androgen deprivation therapy (ADT). Currently, the recurrent disease is controlled well, with regression of the node lesion and a PSA value below the detection level. Two patients were considered to have BCF due to elevated PSA, although no recurrent lesions were identified in the imaging studies; these patients received ADT, and their PSA level was maintained within the undetectable range.

No patient experienced ≥ grade 3 toxicities or newly developed urinary incontinence. Thirty-nine patients had urinary symptoms other than incontinence, which were mild or controlled by medication such as tamsulosin. Within 6 months after the BT, most of these symptoms returned to the baseline level before BT without the need for self-catheterization or catheter indwelling. Four patients had ≤ grade 2 hematuria, which was self-limited. Rectal side effects were observed in eight patients, with transient or intermittent rectal bleeding. Among them, three patients underwent argon plasma laser coagulation for mild rectal proctitis confirmed with colonoscopy, and their symptoms improved after the procedure. In terms of sexual function, of the 71 patients who were potent before BT, 2 patients became impotent and 8 patients experienced weakened erection during the follow-up period.

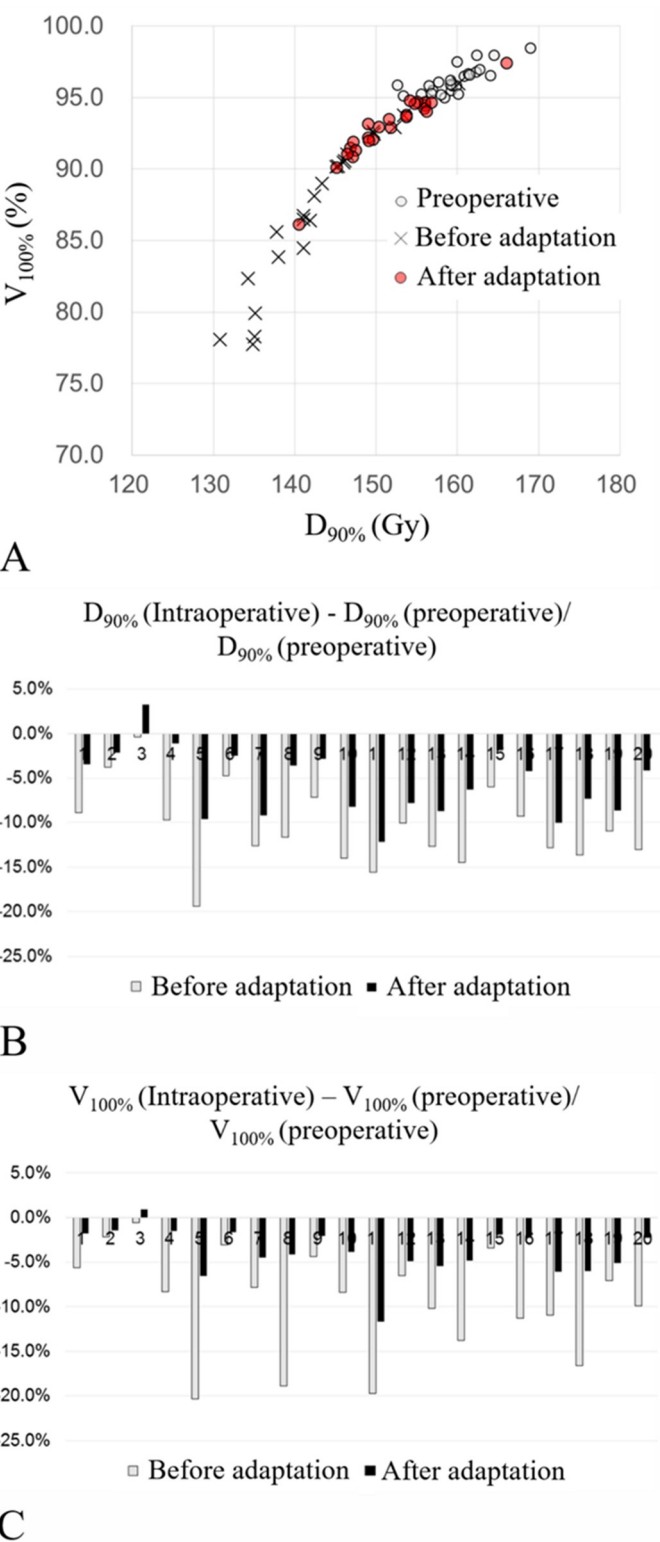

**Fig 5.** (A) D90% and V145 Gy for preoperative planning, before and after intraoperative optimization. (B) The difference between the intraoperative and preoperative plan in D90% was divided by the preoperative D90%. (C) The difference between the intraoperative and preoperative plan in V145 Gy was divided by the preoperative V145 Gy. In (B) and (C), light bars indicate the calculated value before optimization and dark bars indicate the calculated value after optimization.

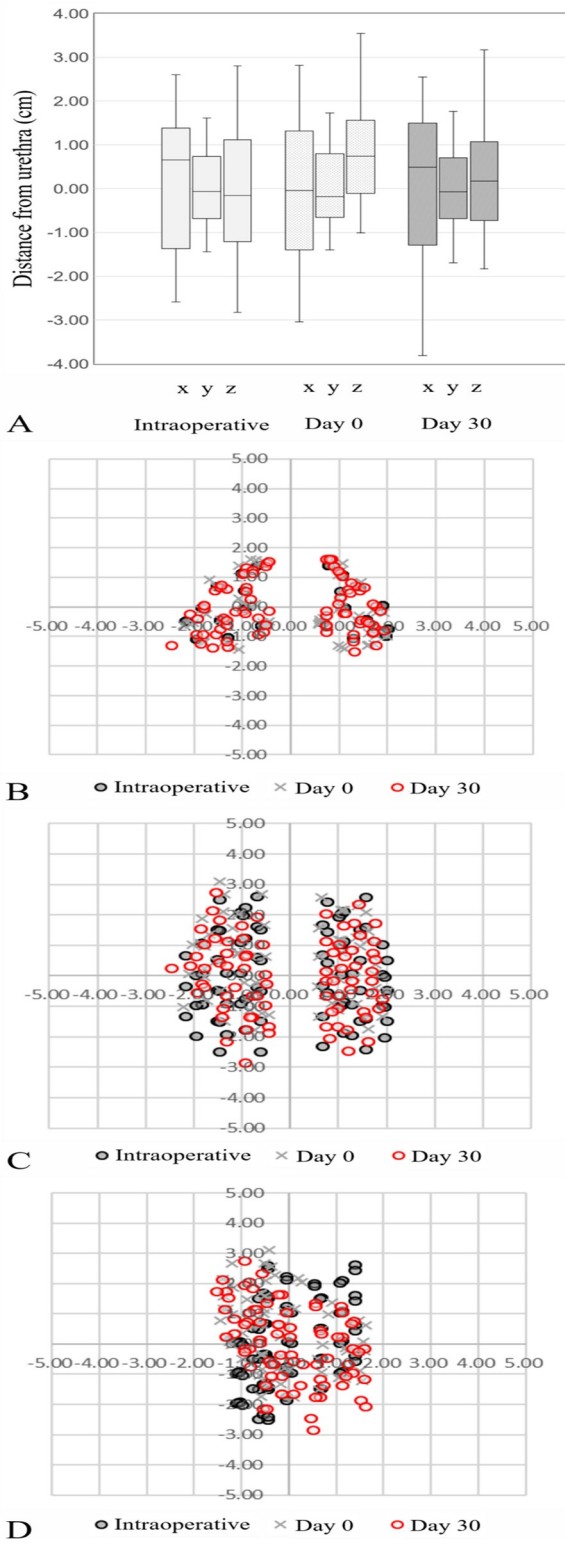

**Fig 6.** (A) Box plots summarizing seeds placement pattern in terms of distance from the urethra in x-, y-, and z-axis. (B), (C), (D) Exemplary intraoperative seed displacement patterns (1 case) on transrectal ultrasound images and post-implant patterns (day 0 and day 30) on computed tomography. The displacement pattern of seeds in the (B) transverse view, (C) coronal view, and (D) sagittal view.

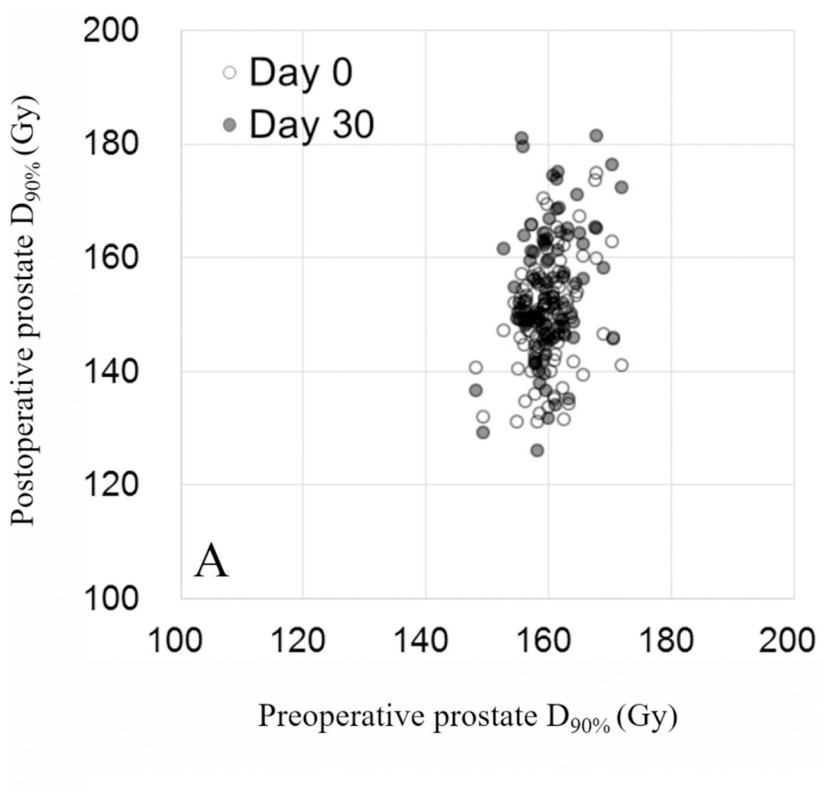

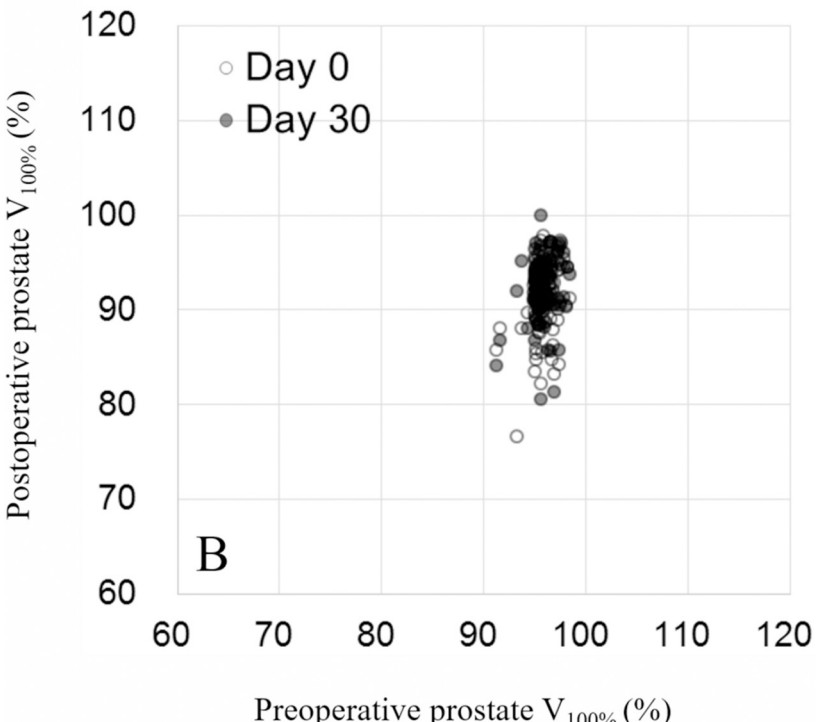

**Fig 7.** Comparison of preoperative planning at day 0 and day 30; (A) prostate $D_{90\%}$ and (B) prostate $V_{100\%}$. These graphs show that all cases were successfully performed to maintain the high dose coverage of prostate from post-implantation day 0 to day 30.

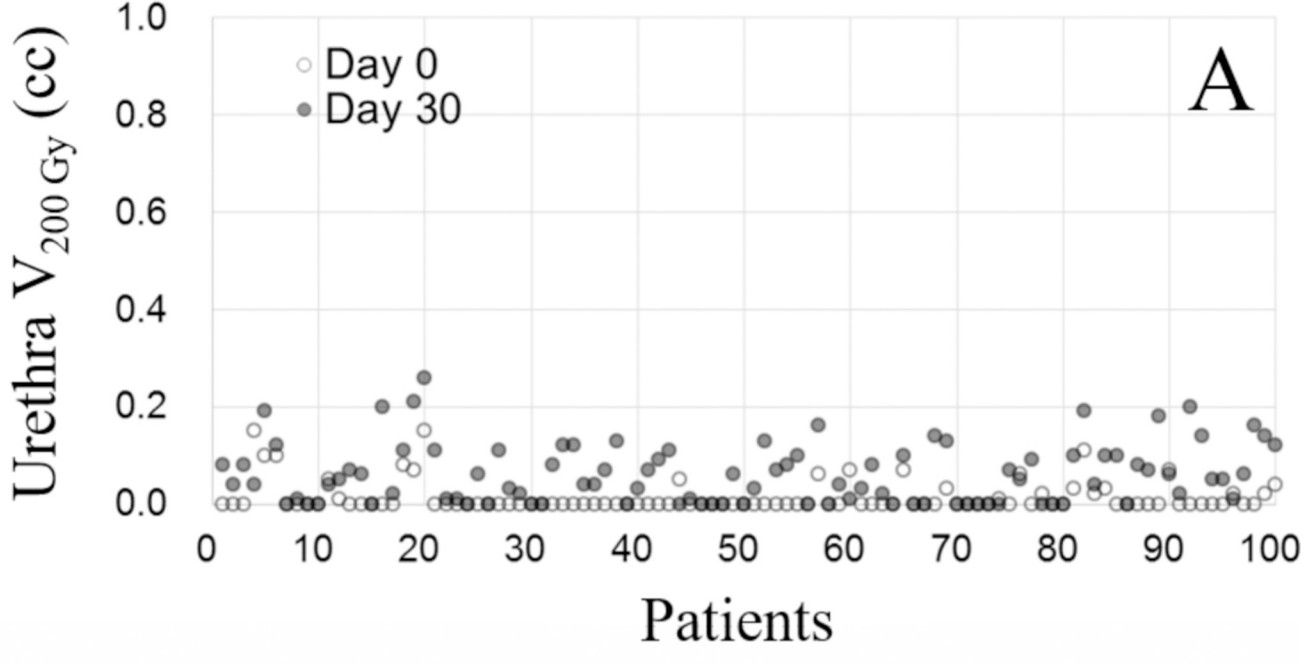

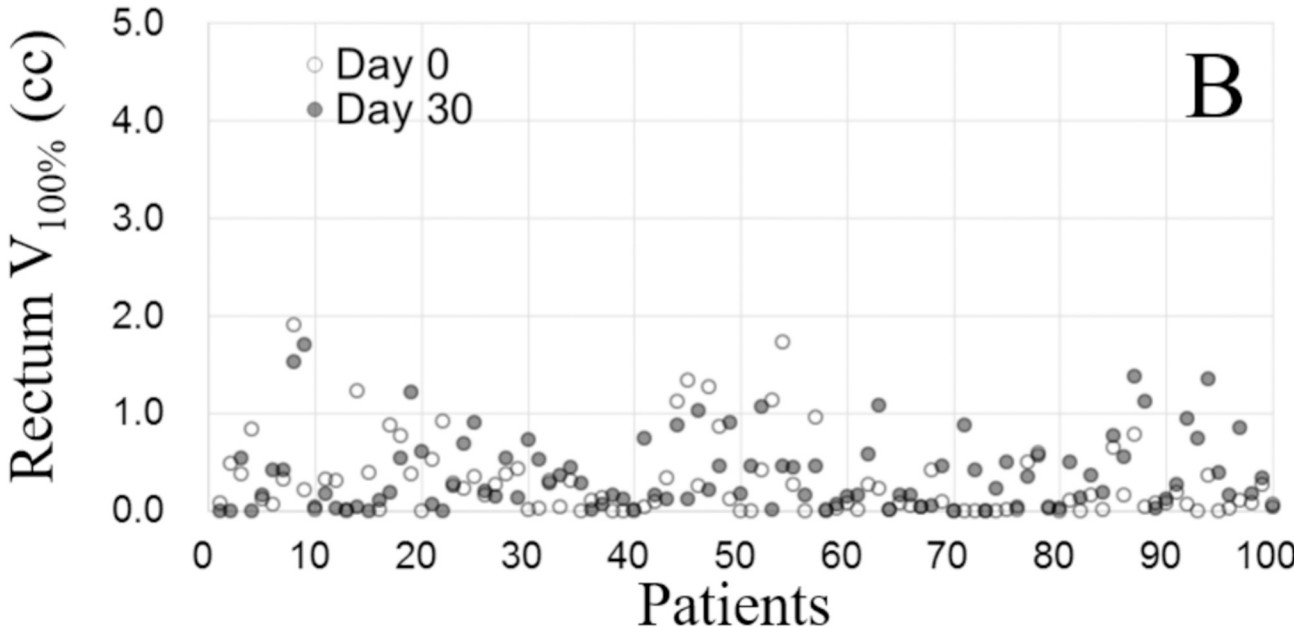

**Fig 8.** Scatter plots showing (A) the urethra V200 Gy and (B) rectum V100%. The values suggest that the doses to the urethra and rectum were maintained at around or lower than the threshold after implantation. In both plots, light dots indicate the values on day 0 and dark dots indicate the values on day 30. Demonstrably, highly controlled implantation was achieved in all cases without excessive dose to the organs at risk.

## Discussion

LDR BT with permanent seed implantation is a safe and effective treatment modality for the management of early-stage prostate cancer. In the treatment of early-stage prostate cancer, both treatment efficacy and treatment-related toxicity are important considerations. BT tends

to induce fewer adverse effects, such as urinary incontinence and erectile dysfunction, compared to surgery [1,3,15–17]. BT also has a relative advantage in terms of dose escalation, as the physical prescription dose and biological effective dose can be up to 2–3 times greater than those in EBRT [18].

In terms of isotope selection, although the difference between isotopes used in LDR BT is not prominent in terms of oncological outcomes [19–21], iodine-125 has a higher average energy and longer half-life than palladium-103 [22], and it requires a lesser number of seeds for BT [23]. Furthermore, notable progression in the techniques of seed implantation, such as the use of stranded seeds and intraoperative real-time dose optimization, has enabled high-quality implantation. The quality of BT depends on both the sustainability of the implanted seeds and proper dose coverage on the target areas, with restricted dose delivery to the OARs. In a prospective randomized comparison between stranded seeds and loose seeds, use of stranded seeds resulted in lesser seed loss [7]. Another study with 1,000 consecutive BT patients who had undergone BT showed a drastic decrease in the seed migration rate from 45.5% in patients using loose seeds to a mere 0.9% in patients using stranded seeds [24]. Likewise, Birckhead et al. reported the outcomes of permanent prostate BT using stranded seeds, with a seed loss rate of 1.0% and seed displacement rate of 0.15%, evaluated using sequential pelvic and chest X-rays acquired at day 0 and 4 months after implantation. In that study, seed displacement was defined as a seed located outside the 1-cm range of a seed cluster [25]. In our study, we did not specifically define seed displacement. However, as shown in the sample case, most seeds maintained their position without significant displacement from the seed cluster. Moreover, no loss of seeds was reported on evaluation of the postoperative CT images, suggesting the sustainability of the current practice using TRUS images and stranded seeds. Given that Kono et al. reported a significant correlation between seed migration and prostate swelling [26], this lack of seed migration might have been correlated to the rarity of profound swelling of the prostate in our patients. An additional advantage is a reduced air kerma strength and the number of seeds required to treat a given target volume. Furthermore, unlike loose seeds, which must be present only in the prostate, the stranded seeds can be located at the prostate capsule or outside; this ensures efficient coverage of the periprostatic fat [27]. When we analyzed the seed location in the preoperative plans in our study, approximately 20–30% of the seeds were located in the periprostatic tissue within the planning target volume (S2 Fig). Combining these properties of iodine-125 isotope and stranded seeds, we could achieve effective dose coverage both within the prostate and in the periprostatic fat without increasing the risk of adverse events.

Precision and sustainability of LDR BT using intraoperative optimization require further evaluation, as previous studies have reported contradictory outcomes. Kudchadker et al. reported outcomes of high-quality implants in prostate BT with preoperative treatment planning and intraoperative optimization. The V100% value after 30 days was 98.6%, and eight of the 100 patients in their study required intraoperative modification [28]. However, another study reported a substantial change in the dose distribution after 1 month, owing to shrinkage of prostate with time and seed displacement [29]. In our study, compared to the V100% values of 95.72% and 92.23% in the preoperative plan and on day 0, respectively, V100% remained stable on day 30, with a median value of 92.18%. Another important dosimetric parameter is D90% for the prostate, as D90% < 130 Gy could be related to an increased risk of recurrence [30]. This parameter was well maintained in our cohort as well, with a median D90% value greater than 150 Gy.

As part of our multiple approaches to achieve high-quality BT, we used MOSFET for in vivo dosimetry to intraoperatively measure the radiation dose to the urethra. The applicability of MOSFET as an in vivo dosimeter during iodine-125 permanent prostate implantation has

been evaluated by a few previous studies. The results of these studies, with a polymethyl methacrylate phantom and small number of patients with intraoperative MOSFET in the urinary catheter, have shown the feasibility of MOSFET as a real-time dosimeter with an acceptable range of error [31,32]. We hereby report our preliminary in vivo measurement data from 50 patients, and to the best of our knowledge, this is one of the largest cohorts with urethral dose measurement using MOSFET. During this initial phase of practice, the efficiency of using MOSFET was uncertain given the effort required. Therefore, its practicality was examined in half of the total patients. Currently, in our institution, MOSFET is routinely used during BT, and the number of cumulative cases has exceeded 200. Real-time urethral dose measurement and appropriate adaptation can ensure minimization of urinary toxicities during BT. Overall, intraoperative optimization with real-time feedback using TRUS images and in vivo dosimetry to evaluate dose delivery to the prostate and OARs enables implantation adjustment given the difference between the preoperative plan and actual implantation.

This study has a few limitations. Implementation of MR imaging could be helpful for the visualization of the prostate and high-risk pertaining region [33,34]. The expense related to this, however, prohibits its routine utilization in clinical practice. Moreover, studies have shown that TRUS and CT could be reasonable alternatives to MR in determining dosimetric parameters [35,36]. From a clinical perspective, it remains controversial whether these parameters related to intraoperative dosimetry could predict or positively correlate with biochemical outcomes after LDR BT. We demonstrated the benefit of intraoperative optimization in improving target coverage while satisfying the dose constraints to OARs in 20 patients in our study. Although intraoperative dynamic planning appears to result in superior biochemical outcomes compared to preoperative planning alone [37], no specific dosimetric parameter has been identified to correlate with biochemical outcomes [38,39]. However, Sasaki et al. reported that there was correspondence between underdosed areas in initial BT and biopsy-positive recurrent sites in patients who experienced BCF with the positive biopsy core [40]. Moreover, there was a case of biochemical disease control in a patient with BCF with negative biopsy results and salvage BT to the area that was underdosed during the initial BT [41]. These outcomes suggest that a cold spot after BT could be a seed of treatment failure, and reducing these spots might be helpful in achieving better results. In this study, we reported good early clinical outcomes after a relatively short median follow-up of 42 months, without prostate cancer- or treatment-related mortality and intractable disease recurrence. However, the PSA kinetics after BT involve a quite unique pathway; up to 80% of patients experience PSA bounce after receiving BT [42], and PSA control requires a long period of time, possibly up to 5 years, before PSA finally reaches the nadir [43]. Therefore, to fully evaluate the complete profile of PSA kinetics and disease control, a longer follow-up period is necessary. Regarding the clinical outcomes of LDR BT with intraoperative optimization, a report solely focused on detailed analysis of BCF-free survival and PSA kinetics is under preparation.

## Conclusions

This study showed that intraoperative optimization can be used to achieve optimal implantation of stranded radioactive seeds in the prostate. Based on intraoperative dynamic planning using TRUS images, intraoperative in vivo dosimetry using MOSFET, and postoperative dosimetry using the CT images acquired on days 0 and 30, we reported the sustainability of seeds within the desired location, with maintained crucial dosimetric parameters, including the V100% and D90% for the prostate, V100% for the rectum, and V200 Gy for the urethra, expecting favorable treatment response.

## Supporting information

**S1 Fig. Linear correlation of total activity between the preoperative and intraoperative plans in cases where additional radioactive seeds were inserted.**
(DOCX)

**S2 Fig. Three-dimensional display of seed distribution relevant to the prostate and adjacent organs.** A few seeds are located within the emerald-colored shade that indicates the planning target volume, constructed by adding a uniform margin around the red-colored prostate.
(DOCX)

## Acknowledgments

We would like to express our sincere gratitude to Dr. Steven J. Frank of MD Anderson Cancer Center for his profound support.

## Author Contributions

**Conceptualization:** Eungman Lee, Jaeho Cho.

**Data curation:** Jason Joon Bock Lee, Dong Wook Kim.

**Formal analysis:** Jason Joon Bock Lee.

**Investigation:** Jason Joon Bock Lee, Tae Hyung Kim.

**Methodology:** Eungman Lee, Jihun Kim, Dong Wook Kim, Jaeho Cho.

**Project administration:** Won Hoon Choi.

**Resources:** Won Hoon Choi.

**Software:** Eungman Lee, Won Hoon Choi, Kyung Hwan Chang, Dong Wook Kim, Han Back Shin.

**Validation:** Tae Hyung Kim, Hwa Kyung Byun.

**Visualization:** Kyung Hwan Chang, Dong Wook Kim, Han Back Shin.

**Writing – original draft:** Jason Joon Bock Lee, Eungman Lee.

**Writing – review & editing:** Jihun Kim, Jaeho Cho.

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
