## [Decision Letter · Decision Letter 0]

22 Nov 2021

PONE-D-21-27735Dosimetric outcomes of intraoperative optimization with adaptive implantation using stranded seeds for low-dose-rate prostate brachytherapyPLOS ONE

Dear Dr. Cho,

Thank you for submitting your manuscript to PLOS ONE. After careful consideration, we feel that it has merit but does not fully meet PLOS ONE’s publication criteria as it currently stands. Therefore, we invite you to submit a revised version of the manuscript that addresses the points raised during the review process.

We look forward to receiving your revised manuscript.

Kind regards,

Jennifer Wei Zou, Ph.D.

Academic Editor

PLOS ONE

Journal Requirements:

2. Thank you for providing a statement regarding patient consent in the Ethics Statement. We ask that you additionally provide this information within the main manuscript.

Reviewers' comments:

Reviewer's Responses to Questions

**Comments to the Author**

1. Is the manuscript technically sound, and do the data support the conclusions?

Reviewer #1: Partly

Reviewer #2: Partly

2. Has the statistical analysis been performed appropriately and rigorously? 

Reviewer #1: No

Reviewer #2: N/A

3. Have the authors made all data underlying the findings in their manuscript fully available?

Reviewer #1: No

Reviewer #2: No

4. Is the manuscript presented in an intelligible fashion and written in standard English?

Reviewer #1: Yes

Reviewer #2: Yes

5. Review Comments to the Author

Reviewer #1: The authors reported an Ultrasound-guided transperineal prostate permanent seed implant LDR monotherapy workflow and the retrospective dosimetric analysis of 100 patient cases undergoing this procedure. Upon reviewing this manuscript, several major flaws are noted, including in presentation and in analysis. I did not find this of adequate quality for publication in its current state.

Major Comments:

1. First and foremost, it is very questionable whether this is "adaptive" planning. Adaptation generally means an optimal plan already exists based on a snapshot at a certain time point (typically sim), and adjustments are made based on anatomical or setup change at time of dose delivery. As I understand, this workflow intentionally uses/encourages a suboptimal (cold) plan/implantation to achieve urethral sparing, and then make up for the coverage intraoperatively with the spare needles (Page 7, Line 125). It is noted that Figure 4B/C shows consistently cold plans "before adaptation". This is just intraoperative optimization with Ultrasound guidance, not "adaptive", because the base plan is intentionally suboptimal to begin with.

Please clarify the workflow, or remove reference to "adaptive/adaptation" in the title and main text.

2. For the reason stated above, Figure 4B/C is very misleading. The before-after isn't really improvement but just fixing what's purposefully done wrong.

3. V100% or V200% are reported in % rather than cc. How is this normalized in post-op, especially in case of prostate volume change. There is no mentioning of this detail.

4. The D90(Intraop) - D90(Preop) metric which the authors refer to throughout the manuscript seems flawed, without an analysis comparing the individual seed positions and those in the preop plan.

5. The concept of intraoperative optimization isn't new. Overall, this work doesn't seem to add much beyond what's reported in Kudchadker et al. doi: 496 10.1016/j.meddos.2012.03.001. The use of intraoperative MOSFET dosimetry is interesting --- is this used to guide optimization? Instead of just reporting the max and mean in-vivo reading (Page 9, Line 167), I recommend adding a detailed plot combing the in-vivo data and TPS dose of the urethra.

6. The workflow needs to be presented much more clearly. Specifically:

a. There is no pre-planning image/dose in the main text at all. Supplementary Figure S2 should be prominently placed in the main text.

b. Page 7 Line 113 says intraop images were compared with sim. How? On the TPS?

c. Page 7 Line 115 says the dose to OARs is evaluated intraoperatively. Is this simply overlay the dose distribution from pre-op plan onto current image? Any registration performed?

d. What is the PTV? Any margins? From the manuscript, it seems to imply the prostate gland itself is the planning volume with no margins. Please clarify.

e. During the intra-operative procedure, are target and OAR contours re-drawn sometimes or always? The manuscript seems to indicate this is case-by-case --- what's the criteria then?

f. Page 7 Line 127 says that kV radiograph is acquired after each "operation". Please clarify the word "operation" --- Is this after the entire OR procedure, or after each needle insertion?

7. First paragraph of Discussion is over-editorializing, comparing LDR and EBRT but ignoring HDR. I recommend removing this paragraph in its entirety as it's unrelated to the results of this work.

Minor Comments:

1. Page 4, Line 49: Remove the extra letters behind [1-3]

2. Supplementary S1. Y-axis. The shorthand for intraoperative is misspelled.

Reviewer #2: This manuscript evaluated prostate LDR plan quality and reported dosimetric outcomes and early clinic outcomes. The study might be beneficial to the clinical application. However, authors should describe the LDR adaptive treatment in more details and it would be better to give some clinical suggestions at the end. Here are my comments and suggestions:

1, what is the dose calculation algorithm used in this study? TG43? What source model was used: point source or line source? As this study reported plan dosimetric characteristic, details about dose calculation should also be reported.

2, MR image was acquired preoperatively and stored in the MiM. Is this only used for prostate volume measurement? This is not included Figure1.

3, in this study, MR, TRUS and CT images were used. It would be better to describe the purpose of these imaging. In addition, multiple software were mentioned like MiM and Variseed. When describe the process, like in line 119, please specified which software was used in each step.

4, more details about planning should be included in the manuscript. Like what technique was used? How to do the optimization? How did the registration help the intraoperative plan?

5, MOSFET was used in this study to verify the urethral dose. How to calibrate MOSFET? As energy of seeds is low compared with external beam radiation therapy. How to get the in-vivo measurement? What is the uncertainty of it? All these details should be included in the paper. Why only 50 patients not 100 patients have in-vivo data? What is the MOSEFT results? Is it consistent with the urethra dose reported from the plan?

6, In abstract, LDR should be included to specify this is LDR prostate study. It would be better to have clinical suggestions in abstract and conclusion.

6. PLOS authors have the option to publish the peer review history of their article (what does this mean?). If published, this will include your full peer review and any attached files.

Reviewer #1: No

Reviewer #2: No

---

## [Decision Letter · Decision Letter 1]

24 Feb 2022

Dosimetric outcomes of preoperative treatment planning with intraoperative optimization using stranded seeds in prostate brachytherapy

PONE-D-21-27735R1

Dear Dr. Cho,

We’re pleased to inform you that your manuscript has been judged scientifically suitable for publication and will be formally accepted for publication once it meets all outstanding technical requirements.

Kind regards,

Jennifer Wei Zou, Ph.D.

Academic Editor

PLOS ONE

---

## [Editor Report · Acceptance letter]

22 Mar 2022

PONE-D-21-27735R1 

Dosimetric outcomes of preoperative treatment planning with intraoperative optimization using stranded seeds in prostate brachytherapy 

Dear Dr. Cho:

I'm pleased to inform you that your manuscript has been deemed suitable for publication in PLOS ONE. Congratulations! Your manuscript is now with our production department. 

Kind regards, 

on behalf of

Dr. Jennifer Wei Zou 

Academic Editor

PLOS ONE